# Vascular Calcification—New Insights into Its Mechanism

**DOI:** 10.3390/ijms21082685

**Published:** 2020-04-13

**Authors:** Sun Joo Lee, In-Kyu Lee, Jae-Han Jeon

**Affiliations:** 1New Drug Development Center, Daegu-Gyeongbuk Medical Innovation Foundation, Daegu 41061, Korea; disjrk@dgmif.re.kr; 2Leading-edge Research Center for Drug Discovery and Development for Diabetes and Metabolic Disease, Kyungpook National University Hospital, Daegu 41404, Korea; leei@knu.ac.kr; 3Department of Internal Medicine, School of Medicine, Kyungpook National University, Daegu 41944, Korea

**Keywords:** autophagy, chronic kidney disease, endoplasmic reticulum, hyperphosphatemia, inflammation, matrix vesicle, mitochondrial dysfunction, osteogenic differentiation, vascular calcification, vascular smooth muscle cell

## Abstract

Vascular calcification (VC), which is categorized by intimal and medial calcification, depending on the site(s) involved within the vessel, is closely related to cardiovascular disease. Specifically, medial calcification is prevalent in certain medical situations, including chronic kidney disease and diabetes. The past few decades have seen extensive research into VC, revealing that the mechanism of VC is not merely a consequence of a high-phosphorous and -calcium milieu, but also occurs via delicate and well-organized biologic processes, including an imbalance between osteochondrogenic signaling and anticalcific events. In addition to traditionally established osteogenic signaling, dysfunctional calcium homeostasis is prerequisite in the development of VC. Moreover, loss of defensive mechanisms, by microorganelle dysfunction, including hyper-fragmented mitochondria, mitochondrial oxidative stress, defective autophagy or mitophagy, and endoplasmic reticulum (ER) stress, may all contribute to VC. To facilitate the understanding of vascular calcification, across any number of bioscientific disciplines, we provide this review of a detailed updated molecular mechanism of VC. This encompasses a vascular smooth muscle phenotypic of osteogenic differentiation, and multiple signaling pathways of VC induction, including the roles of inflammation and cellular microorganelle genesis.

## 1. Introduction

Vascular calcification (VC) is defined as mineral deposition, in the vasculature, in a form of calcium-phosphate complexes. Although VC is regarded as part of the normal aging process, certain pathological processes such as diabetes, hypertension, chronic kidney disease (CKD), and or rare hereditary disorders, may also precipitate the condition [1].

Traditionally, calcification is classified into two forms, depending on where the mineral is deposited. Intimal calcification is closely related to lipid deposits, and the clinically relevant infiltration of inflammatory cells, with obstructive arterial disease, whereas the latter is more pronounced by transformation into osteoblast-like cells from smooth muscle cells. Medial calcification is more prevalent in the aforementioned diseases, including CKD, diabetes, and arterial stiffness, rather than obstructions that are clinically significant in this pathology [2].

VC has an inseparable relationship with atherosclerotic vascular disease [3]. In addition, it is known that arterial stiffness, which represents the functional disturbance of VC, is an independent predictor of cardiovascular mortality [4]. Loss of elastin is coupled with medial calcification, and the degradation of elastin is believed to further contribute to the osteogenic process in aortic tissue [5]. Elastin is the most abundant protein in the walls of the arteries, and thus its loss increases medial calcification, increased pulse pressure, left ventricular hypertrophy, and eventually, cardiovascular death [3]. A recent report showed that monitoring of 10-year ambulatory blood pressure changes in older hypertensives revealed that 24-h pulse pressure better predicts mortality than 24-h systolic blood pressure [6]. This finding suggests that in clinical perspective, reduction of arterial stiffness should be underscored as much as lowering systolic blood pressure.

In addition, despite continuous attempts, caclciphylaxis and ectopic calcification remains a clinical unmet need in certain clinical settings. As a result of recent advances in the field of VC pathogenesis, our current understanding of the mechanism of VC encompasses chronic inflammation, autophagy defects, endoplasmic reticulum (ER) stress, and mitochondrial dysfunction and its dynamics.

In this review, we summarize traditionally well-known mechanisms of VC, followed by recent updates regarding VC pathogenesis.

## 2. Classification of Vascular Calcification, Depending on Site and Etiology

VC is caused by the deposition of pathological mineral in the vascular system. Calcification has been classified, depending on where the mineral was deposited. VC of the vessel wall occurs in the intimal and medial layers. It can also be found in valvular calcification and calciphlaxis. Intimal calcification is associated with atherosclerotic plaques and is known as the result of lipid accumulation, macrophage invasion, proliferation of smooth muscle cells, and dysfunction of extracellular matrix proteins, in response to chronic arterial inflammation [7]. These unstable and rupturable plaques form on the inner blood vessel wall, causing obstructive vascular disease.

In atherosclerotic neointima, calcification is initiated as microcalcifications, which, when present in the vascular wall, might play a pivotal role in overt atherosclerotic plaques [8]. Microcalcification is defined as small calcium deposit (< 5 µm), initiated by necrotic or apoptotic cell death within lipid core. This type of calcification is observed in high-risk plaque. If the rupture of the plaque is perturbed due to the subsequent thrombotic occlusion, healing of the necrotic core and subsequent formation of macrocalcification (> 5 µm) starts, resulting in plaque stabilization [9].

This process is believed to be derived from apoptotic smooth muscle cells (SMCs), or matrix vesicles (“exosomes”) released by these cells near the internal elastic lamina. This occurs before changes in the intimal content of calcification-regulation proteins, such as osteocalcin (OC) and bone morphogenic protein (BMP) 2, but coincided with enhanced expression of uncarboxylated matrix gla protein (MGP) [7]. Increased nucleation sites may facilitate the precipitation of Ca^2+^ salts at the micrometer scale. Early-stage microcalcifications play a pathological role in the onset and progression of vascular disease.

Medial calcification is associated with aging, diabetes mellitus, hypertension, osteoporosis, and CKD [10]. It can occur even without vascular stenosis, which mainly causes vascular stiffness, and increases the incidence of cardiovascular complications [11,12,13]. The medial layer of the vessel wall is composed of smooth muscle cells and the elastin-rich extracellular matrix. In medial calcification, the process of differentiation of smooth muscle cells (SMCs) into osteoblast-like cells is akin to bone formation, and is related to genes such as *BMP2*, *Msh Homeobox 2* (*MSX2*), and alkaline phosphatase (*ALP*) [14,15,16].

It is also known that this differentiation begins with calcium deposition, through the production of matrix vesicles produced by SMCs [17]. This phenomenon becomes possible by reducing calcification inhibitors, increasing oxidant or endoplasmic reticulum (ER) stress, during SMC signaling, apoptosis, and disorder of calcium-phosphate homeostasis associated with DNA damage, resulting from abnormal system hormonal regulation [18].

Arterial stiffness and medial calcification are independent predictors of cardiovascular mortality, and have been strongly and reciprocally correlated in many clinical studies. Each process intensifies each other, to create a vicious cycle [19]. Here, hypertension plays a critical role in this vicious cycle [20]. Hypertension promotes extracellular remodeling by accelerating type 1 collagen, fibronectin, and proteoglycans accumulation primarily in intima media [21]. In addition, aberrant elastin production and deposition are accompanied [22,23]. Newly synthesized fibers under pathological stimuli are less effective, therefore qualitative abnormalities of elastin occur [24]. In addition, elastin-rich extracellular matrix (ECM) is degraded and causes accumulation of elastin-derived peptides [24]. This cascade licenses a phenotypic switch of SMC. SMC endowed with proliferative and migratory activity plays a critical role of osteogenic differentiation of vascular cell [21,24]. This pathological environment induces VC which further accelerates vascular stiffness and increase of pulse pressure [21]. This phenomenon is observed in clinical situation. Compared with normotensive patient, patients with isolated systolic hypertension manifested with increased calcification of abdominal and descending thoracic aorta [25].

### 2.1. Valvular Calcification

Aortic valvular calcification, aortic sclerosis, and aortic stenosis, seen in different stages of calcific aortic valvular disease share a common pathophysiological process [26]. The main cause of calcific aortic valvular disease is age-related degenerative process [27]. In addition, genetic predisposition [28,29,30] such as *IL-10* [31], *Lp(a)* [28], or *apolipoprotein B* [32] polymorphism or congenital valvular defects [33] can cause valvular calcification.

Traditional risk factors of atherosclerosis as arterial hypertension, kidney failure, male sex, diabetes, and dyslipidemia are also regarded as a risk factor of early-onset valvular calcification [34]. When severe valvular calcification results in aortic valve narrowing, it is called aortic stenosis. If the aortic valve becomes narrowed and severely dysfunctional, replacement surgery is required [26,35]. In this case, aortic calcification can be seen as an early sign of heart disease, thus preemptive efforts to prevent the development of serious conditions by inhibition of calcification are required.

### 2.2. Calciphylaxsis

Calciphylaxis is a clinical resultant syndrome of arteriolar calcification, commonly investigated in end-stage renal disease patients on dialysis [36]. It is induced by intense deposition of calcium accompanied by intimal proliferation, fibrosis, and thrombosis [37,38,39,40]. These processes eventually lead to necrosis or ischemia in small blood vessels, skin, and other organs [41]. Risk factors for calciphylaxis are high calcium-phosphate product [42], elevated level of parathyroid hormone [43,44], hypoalbuminemia [45,46], diabetes [46,47,48], female sex [45,49], obesity [50], and warfarin overdose [51,52]. This disease is rare, but fatal and even if it is diagnosed at an early stage, the mortality rate is exceptionally high and the success rate of healing is low [53]. The exact cause of calciphylaxis is still unknown, but its pathology includes tunica medial calcification, necrosis of tissue. It is common in ESRD patients, therefore it is likely that intimal hyperplasia and medial calcification is entangled with etiology [54].

## 3. Vascular Smooth Muscle Cell Phenotypic Differentiation in High-Phosphate Environments

Disruption of mineral homeostasis and high phosphate levels are considered to be the main determinants of VC in CKD [14,55]. Hyperphosphatemia often occurs as a result of renal failure [55]. However, the effect on calcification of the phosphate binder is not apparent [56]. This is presumably due to the intracellular system that regulates phosphate, and the availability of bone to supply phosphate.

It is well accepted that phosphate complexes activate pro-calcific intracellular signaling pathways [57,58]. Increased levels of calcium phosphate products associate with the development of vascular calcification in CKD [59]. Even when renal function is normal, increased phosphate in serum associates with cardiovascular mortality and coronary artery calcification, suggesting that phosphate plays an important role in the pathophysiology of VC [60].

The mechanism of initiation and progression of VC is similar to the phenomenon of physiological bone formation [61]. High phosphate upregulates *Pit1* to raise intracellular levels of inorganic phosphate (Pi), inducing runt-related transcription factor 2 (*RUNX2)*. This enhances the osteogenic transition of vascular SMC (VSMC) [62] (Figure 1). Downregulation of calcification inhibitors, release of extracellular vesicles, and remodeling of extracellular matrix, as well as osteo-/chondrogenic transdifferentiation and apoptosis of vascular cells contribute to this pathology. VSMCs play a key role in this phenomenon, and the processes are not mutually exclusive [59,63].

Calcium and phosphate concentrations exceed their solubility under pathological conditions, and endogenous calcification inhibitors are required to prevent ectopic precipitation of calcium and phosphate [64]. High extracellular phosphate levels inhibit the production of calcification inhibitors, and promote the production of exosomal vesicles lacking such inhibitors [65,66].

Hyperphosphatemia also induces extracellular matrix remodeling. The production of matrix metalloproteinases and cysteine proteases results in the degradation of matrix proteins, generation of bioactive elastin peptides, and the synthesis of collagen, creating a collagen-enriched ECM [67,68,69]. In addition, hyperphosphatemia induces the expression of enzymes that regulate collagen crosslinking, and supramolecular organization [70]. These events combine to cause extracellular matrix remodeling, which is involved in vascular mineralization.

Under high-phosphate circumstances, apoptosis or necrosis of VSMCs is induced [14,68,71,72], possibly generating apoptotic bodies from VSMCs, which could serve as nidi for calcium phosphate precipitation [71,73]. Phosphate-regulated intracellular signaling includes Wnt/β-catenin, protein kinase B (PKB or Akt), nuclear factor-kappa B (NF-κB), and serum- and glucocorticoid-inducible kinase 1 (SGK1) [57,73,74,75,76]. The role of the inflammasome is also important [77,78]. In addition, oxidative stress, caused by an imbalance of antioxidants and reactive oxygen species (ROS), is related to osteoinductive signals and apoptosis, and may be regulated by the Gas6/Axl pathway, Akt, AMP-activated protein kinase (AMPK), etc., in which phosphate directly regulates apoptosis [77,79,80].

In addition, it is known that vascular calcification affects intracellular signaling, due to epigenetic phenomena, including altered microRNA levels, DNA methylation, and histone modifications. In addition, aging affects calcification through osteoinductive signaling, inflammatory cytokines, and oxidative stress [78,81,82,83,84,85]. Moreover, cellular phenomena that affect osteoinductive intracellular signaling, via hyperphosphatemia, include autophagy, ER stress, and mitochondrial dysfunction. Eventually, various signaling pathways trigger phosphate-induced osteo-/chondrogenic transdifferentiation of VSMCs, as associated with VC [86,87,88]. These complex and cross-talking mechanisms closely support the postulate that VC is affected by phosphate levels. Finding critical pathways among these will be an important approach in treating VC [59].

In addition to changing the properties of VSMCs, formation of substrates that can cause calcium deposition, and the alteration of defenses to remove them, are known to be important factors in initiating or worsening calcification [59]. Representative causes of deposition include apoptosis, autophagy inhibition, matrix vesicle production, and increased bone mineralization, due to increased bone resorption.

## 4. Main Factors Causing Vascular Calcification

### 4.1. Extracellular Vesicles

Extracellular vesicles are secreted, membrane-enclosed particles, such as microvesicles, matrix vesicles, multivesicular bodies, ectosomes, exosomes, microparticles, and apoptotic bodies [89]. The inner core of extracellular vesicles consists of soluble proteins, lipids, and noncoding RNAs. Based on their mechanism of biogenesis, extracellular vesicles are classified into three types: exosomes, microvesicles and apoptotic bodies [90]. Extracellular vesicles can transfer functional transcripts and lipids to target cells, triggering changes in the target cell’s phenotype [91]. EVs secreted into body fluids, and microenvironments from other cell types, act in adjacent regions or in distant cells [92].

Extracellular vesicles are found in calcified human aortic valves, aortic media, and atherosclerotic intimal plaque [93]. Pathological extracellular vesicles are secreted by SMCs, stromal cells, and macrophages, all in vessel walls [94]. In the normal state, contractile VSMCs release extracellular vesicles, especially matrix vesicles, to maintain homeostasis [95]. MVs contain vitamin K-dependent matrix GLA protein (MGP) and circulating fetuin-A (Fet-A) [95]. However, under pathological states, VSMCs transform into a synthetic phenotype to promote the secretion of matrix vesicles, and transform target cells into a calcified state [93]. Moreover, SMC-derived calcifying extracellular vesicles aggregate and form microcalcifications, themselves [94]. Large calcifications are formed by the accumulation of microcalcifications, and maturation of minerals.

Collagen acts as a scaffold in controlling size and shape during this growth process [96]. However, in pathological environments (e.g., CKD (hyperphosphatemia), inflammation-driven atherosclerosis), VSMCs release extracellular vesicles, which contain more calcification-associated markers, and less calcification inhibitors [97]. In atherosclerotic plaques, collagen receptor-deficient VSMCs show increased release of calcified extracellular vesicles, and precipitation of collagen and minerals [97]. In this situation, transported proteins, in calcified EVs, possess the ability to uptake Ca^2+^ and inhibit Fet-A activity [66]. Calcified extracellular vesicles may then induce osteogenic genes (*RUNX2*, *SMAD1*, osterix (*SP7*), tissue non-specific alkaline phosphatase (*TNAP*), and proinflammatory genes [59,98].

It has been suggested that VC, in CKD, associates with increased deposition of VSMC-derived vesicles [92,99]. In one study, using electron microscopy, matrix vesicles contained calcium phosphate crystals that were deposited in the ECM, in patients with predialysis or dialysis, whereas MVs were not observed in healthy individuals [100].

Calcium-carrying EVs are regarded as an adaptive reaction because they extrude calcium from the cell, to prevent excess accumulation of intracellular calcium [101]. In response to calcium overload, or calcifying stimuli, EV release initially occurs until eventually, the deletion of the calcification inhibitor causes the EV to change into calcified extracellular vesicles [96]. Moreover, inhibition of sphingomyelin phosphodiesterase 3 (SMPD3), an important player in extracellular vesicles production, prevents extracellular vesicles secretion and VSMC calcification, suggesting that extracellular vesicles could be a therapeutic target for VC [97].

### 4.2. Endoplasmic Reticulum (ER) Stress

Protein folding and maturation, during the process of protein synthesis, occurs in the ER, the first organelle of the secretory pathway [102]. ER stress is induced when the folding requirements of a protein exceed the ability of the ER to process it, or in the loss of calcium homeostasis [103]. Additionally, ER stress is related to B cell differentiation into plasma cells, insulin secretion by pancreatic β-cells, and osteoblast maturation, during bone formation [104].

Since bone formation requires secretion of large amounts of ECM and regulatory proteins, it has been suggested that increasing protein-folding capacity, and activation of bone-specific gene transcription, via signaling during the unfolded protein response (UPR) [105]. VCs and bone formation share similar genes, and calcification inhibitors, in terms of the UPR, and the interrelationship between these processes, have been studied [104]. The role of three branches of the UPR, in VC, is as follows.

VC, which is induced in rat models by nicotine and vitamin D, is accompanied by upregulation of chaperone gene (*Grp78*, *Grp94*) expression, and ER stress-induced apoptosis markers (*CHOP*, *CASP12*) in the affected aorta [106]. Since ATF4 is also considered a mediator of ER stress-induced calcification [107], a PERK-eIF2a-ATF4-CHOP complex forms and is upregulated in animal models of VC (e.g., the *ApoE*^−/−^ mouse, 5/6 nephrectomy, etc.) [108,109]. Therefore, in addition to CHOP, ATF4 is suggested to be required for VSMC mineralization [109].

In vitro models suggest that stearate-induced calcification in MOVAS-1-murine aortic VSMCs, occurs via PERK pathway signaling, and anomalous *XBP1* splicing [110]. In addition, ATF4 regulates osteogenic differentiation and mineralization in the same model. In human VSMCs, BMP2 mediates oxidative stress-induced ER stress, increasing signaling by the IRE-1-XBP1-GRP78 pathway [111]. ER stress also increased expression of the transcription factor XBP-1, which binds to the *RUNX2* promoter, effecting VSMC differentiation and calcification [111].

### 4.3. Autophagy Inhibition

Autophagy is required for removal of misfolded proteins, damaged organelles, or unwanted metabolites [112]. The autophagosome fuses with lysosomes, and their engulfed contents are degraded into biosynthetic biomass, and also recycled to provide cellular energy [113]. Autophagy is a critical pathway for maintaining normal VSMC function [114]. Autophagy is also classified into selective and nonselective pathways, with the latter degrading cytoplasmic contents. The selective pathway, by contrast, degrades specific substrates, using specific autophagy receptors [115,116,117]. Under normal physiologic homeostasis, low levels of basal autophagy are always present to maintain proper protein turnover, and recycling of damaged organelles [118]. However, when faced with stress (e.g., nutrient starvation, excessive ROS, hypoxia, infection, hyperphosphatemia, protein aggregation, etc.), the autophagy pathway is upregulated. Hyperphosphatemia is well known to induce calcification by promoting osteogenic transformation of VSMCs, and dysfunction of endothelial cells, through activating autophagy [88]. High concentrations of inorganic phosphate (Pi) enhance autophagic flux, by suppressing mammalian target of rapamycin (mTOR) signaling [119]. This is indicative of protective role of autophagy in endothelial cells, against Pi-induced apoptosis.

Recently, increased expression of LC3 was observed in endothelial cells of a CKD rat model, compared to sham-operated controls. By contrast, in vitro models show that autophagy plays a protective role, to counteract against ROS-induced VC, at high Pi concentrations [88]. Increased autophagy, as evidenced by induction of LC3-ll, p62, lgfbp3, and Atg16l1, in aortic VSMCs, was noted in a uremic media calcification model, in DBA/2 mice fed a high Pi diet [120].

It is well known that autophagy is related to endothelial cell homeostasis, phenotype transition, and VSMC calcium homeostasis [121,122,123]. In particular, platelet-derived growth factor (PDGF) is a crucial phenotype-switching cytokine that is upregulated during stressful conditions (e.g., hypertension, atherosclerosis, diabetes, injury, etc.) [124]. Recent studies show that PDGF stimulates cell phenotype switching, while activating autophagy. Activated autophagy induces degradation of proteins, including α-smooth-muscle actin (SMA), calponin, and SM22a, proteins required to maintain the contractile phenotype [125].

Hyperglycemia is also a well-known inducer of VC. In this process, O-linked N-acetylglucosamine (O-GlcNAcylation) induces protein modifications in diabetic arteries [119]. Here, O-GlcNAcylation of Akt leads to its activation, which results in VC. Activated Akt induces mTOR activation, leading to suppressed autophagy, while inhibition of mTOR, by rapamycin, ameliorated VC induction, supporting the possibility of a novel mechanistic role of autophagy, in hyperglycemia-related VC [119].

The aforementioned MVs, released by VSMCs, represent another main factor in the etiology of VC. Those vesicles are enriched with Ca^2+^ and phosphate (Pi), along with TNAP, ecto-nucleotid pyrophosphatases/phosphodiesterase 1 (ENPP1), Na^+^/K^+^ ATPase, PHOSPHO1, and Pit1 [126]. Importantly, MV membranes are highly enriched with the phospholipid phosphatidylethanolamine which is a major component of the autophagosome membrane [126,127,128].

In addition, calcification precursors such as calcium and phosphate are formed or processed in the endosome, multivesicular bodies, autophagosomes or autolysosomes, and eventually taking part in the regulation of intracellular calcium and phosphate homeostasis [129]. Indeed, calcium phosphate precipitates that are used in DNA transfection localize within autophagosomes, which also support this phenomenon. During the autophagic process, LC3-positive vesicles colocalize with ubiquitin and p62, a selected autophagy adaptor, and then colocalize with the lysosomal marker LAMP1, to eventually contribute to the completion of the autophagy cycle [130]. During the process, calcium and phosphate are mobilized and processed accordingly, and following lysosomal lysis, the precipitates are utilized as precursors of VC. Calcified hydroxyapatite, which is packed into autophagosomes, during their formation, have been confirmed in calcified mouse primary osteoblasts [131].

Based on previous reports, modulating autophagy was investigated as a preventive strategy against VC. Likewise, representative autophagy inducers, such as rapamycin and valproic acid, have been shown to inhibit VSMC calcification in vitro [120,132]. However, since there is a possibility that pharmaceutical manipulation exhibits a nonspecific, off-target effect, it is necessary to study the therapeutic effect on calcification, as a method to determine autophagy-specific effects.

### 4.4. Apoptosis

Apoptosis is one of the pathways involving inorganic phosphate (Pi) uptake, via a sodium-dependent phosphate cotransporter (Pit-1) [62]. In particular, VSMC apoptosis significantly contributes to vascular calcification by Pi [133]. Thus, VC is initiated in the presence of apoptotic bodies, and matrix vesicles, which are generated from apoptotic and viable VSMCs that serve as a nidus of calcium phosphate deposition [134], and moreover, the Gas6/Axl/Akt pathway has been reported as a damage survival pathway, by Pi uptake [79].

Mitochondrial dysfunction also induces apoptosis by Pi uptake [79]. In turn, induced apoptosis generates oxidative stress, further mitochondrial dysfunction, apoptotic bodies, and MVs [135,136,137,138], events that have been mechanistically linked to DNA damage, ER stress, autophagy or mitophagy, and calcium phosphate deposition. It can also induce osteogenic gene induction and transformation into osteogenic phenotype of VSMCs [139,140,141,142]. Apoptosis is not mutually exclusive, but actually complementary to other pathways introduced in this review that eventually lead to VC.

### 4.5. Osteoporosis

VC is observed in a severely advanced form of cardiovascular disease (CVD) [143], which is now considered an active regulated process, similar to the bone formation process, especially in terms of mineral composition, initiative metabolism, developmental processes, and gene expression. The mechanism by which calcification occurs is similar to bone formation, but the environment is somewhat more similar to bone resorption [144]. The increase in minerals in the blood, following bone loss, may play a role in initiating calcification or worsening the phenomenon. Previous reports suggest that bone loss may promote CVD [145]. VC is a representative link between bone loss and CVD, and involving similar pathological conditions such as high phosphate and calcium levels, hyperparathyroidism, low levels of calcification inhibitors, and coexistence of hypertension and atherosclerosis [56]. Strategies for treating disease manifesting low-bone mineral density basically include preserving bone formation, while focusing on the reduction of bone resorption. Extraosseous calcification shares several common molecular mechanisms with the process of bone formation. Therefore, directly targeting VC may cause unwanted reduction of bone formation [146].

Due to these considerations, a strategy for treating osteoporosis (amelioration of excessive bone resorption), rather than targeting osteogenic signaling, could be a better direction for treating VC. The osteogenic factors BMP2 and osteopontin (OPN) are both induced by calcium phosphate nanocrystals, rather than soluble phosphate [147]. Moreover, extracellular pyrophosphate is well known as a potent inhibitor of VC [148], and parathyroid hormone (PTH) and 1,25-dihydroxyvitamin D (1,25-(OH)_2_D) are main regulators of circulating levels of calcium and phosphate [149]. Although PTH metabolism is complex, and depends on the environment [150], vitamin D stimulates the absorption of calcium and phosphorus, followed by promoting osteoblast differentiation and the expression of different bone modulators [151].

### 4.6. Lipoprotein(a)

Lipoprotein(a) (Lp(a)) is a lipoprotein particle, is characterized that the spanning of apoB-100 protein bond with glycoprotein, apolipoprotein [152]. Its genomic variant is strongly correlated with the existence of aortic valve calcification and stenosis.

Its level is positively correlated with presence of coronary calcification and aortic valve calcification [153]. Lp(a) also has a potential role as a carrier for pro-calcific and pro-inflammatory factors. When the endothelial injury (mechanically stressed aortic valve leaflets or atherosclerotic plaque) is induced, Lp(a) carries oxidized phospholipids, enzyme, autotaxin to injury area. These features can enhance the calcification process in these lesions. Recently identified role of Lp(a) during aortic valve stenosis involves NF-κB cascade that induces *interleukin (IL)-6*, *BMP2* and *RUNX2* [154]. In addition, monoclonal antibody against to oxidized phospholipids attenuates osteogenic differentiation by Lp(a) in vitro [155]. These findings suggest reducing Lp(a) or oxidized phospholipids can be a therapeutic strategy for delay of arterial valve stenosis or replacement.

### 4.7. Osteogenic Markers in VC and Osteoporosis

Estrogen inhibits VC by modulating the receptor activator of nuclear factor-kappa B (RANK) and RANK ligand (RANKL) signaling pathway [156], and another metabolic hormone, leptin, promotes osteogenic differentiation and VC [157]. Adiponectin, however, reduces VC processes, through inhibition of ER stress and reduced osteoblast differentiation [158].

The Prospective Epidemiological Risk Factors Study Group, a 7.3-year study of follow-ups of postmenopausal men and women, showed that aortic calcification associated with lower bone mineral density (BMD), and bone loss, from the proximal femur [159]. Likewise, the Multi-Ethnic Study of Atherosclerosis’s (MESA’s) Abdominal Aortic Calcium Study (AACS) showed that low BMD strongly associates with increased deposition of coronary artery calcium [160].

Bone biomarkers are likely the key factors in both physiological bone formation and the pathogenesis of disease states, including VC (Figure 2). Several factors are listed in the following sections.

#### 4.7.1. Inducers of VC

##### Cathepsin K

Cathepsin K is a major lysosomal cysteine protease that degrades organic bone matrix in osteoclasts [2,161]. Cathepsin K is predominantly expressed in osteoclasts, and is increased during VC, as well as in osteoporosis and CVD [162,163]. Recent research indicates that deletion of cathepsin K ameliorates VSMC calcification, by diminishing VSMC differentiation, and blocking Wnt3a and osteoprotegerin (OPG) pathways. Therefore, targeting cathepsin K is a promising strategy for the prevention of VC.

##### Fibroblast Growth Factor 23 (FGF23)-Klotho

Fibroblast Growth Factor 23 (FGF23) is predominantly produced by osteocytes, in response to phosphate levels [164]. It acts in collaboration with the transmembrane protein Klotho as a cofactor, to reduce phosphate resorption in the proximal renal tubule, and decrease intestinal phosphorus absorption, by reducing synthesis of calcitriol in the kidney [165,166]. These functions are available only in the presence of klotho. In the absence of klotho, even high concentration of circulating FGF23 cannot regulate systemic phosphate homeostasis [167]. As a result that serum FGF23 levels increase in response to high phosphorous in a compensatory manner, these rising levels can be considered as an indirect surrogate marker of VC.

FGF23 reduces 1α-hydroxylase activity, which decreases the conversion of calcidiol into its active form 1,25-(OH)_2_D, while upregulating 24-hydroxylase, which breaks down vitamin D [165]. Thus, serum FGF-23 positively correlates with PTH and phosphate levels, and negatively correlates with 1,25-(OH)_2_D, glomerular filtration rate, and tubular phosphate reabsorption [168]. FGF23 levels also increase during the initiation stages of CKD, and plays a key role in mineral ion changes and bone metabolic disorder [169]. FGF23-deficient mice show hyperphosphatemia, hypercalcemia, high 1,25(OH)_2_D, and reduced PTH levels, conjointly indicating defective skeletal mineralization [169].

FGF23 is also elevated when renal function declines, and peaks in end-stage renal disease. It is also used to predict morality of CKD patients [170]. These findings suggest FGF23, in conjunction with Klotho, is implicated in osseous as well as extraosseous (soft tissue) mineralization processes.

##### Bone Morphogenic Protein (BMP)2

BMPs act by binding to a heterodimeric complex of transmembrane receptors (BMP receptors l and ll) [171]. BMP, which exists in various subtypes, signaling is activated through phosphorylation and nuclear translocation of Smad [172]. Among the subtypes, BMP2 is best known for its role in the development of VC [173]. Numerous studies have demonstrated that BMP2 is involved in VC [174], since it contributes to the transdifferentiation of VSMCs into osteochondrogenic cells [175], and BMP2, combined with BMP4, localizes to sites of VC [176]. BMP-2 and BMP-4 have also been most associated with calcific arteriopathy [177], although BMP7 might exert inhibitory effects against VC [176]. In bone formation, both BMP2 and BMP7 induce osteoblast differentiation, through increased expression of runt-related transcription factor 2 (Runx2) and osterix [178,179,180].

BMP2 suppresses VSMC proliferation through p21 inhibition, followed by cell cycle arrest [181,182]. Intimal hyperplasia, induced by balloon injury, is reduced by BMP2 [183], which also effects loss of SMC markers, and acquisition of osteoblastic profile gene expression, including *Msx2*, *alkaline phosphatase(ALP)*, *OPN*, and others [171,184,185,186]. Furthermore, BMP2 elevates apoptosis, production of ROS, and inflammation in VSMCs [59,187].

Consequently, BMP2 is a strong basic causative factor for VC, which is opposed by BMP7. Several underlying mechanisms have been suggested, including specific receptor differences, and the existence of Smad-independent pathway.

##### Alkaline Phosphatase (ALP)

Alkaline phosphatase (ALP) is a phenotypic marker of bone formation and VC, for which it is considered essential [188,189]. ALP activity is important for hydroxyapatite formation, during endochondral ossification, and a similar role in the vasculature has been observed in ectopic calcification, wherein ALP degrades PPi.

##### Runx2

Runx2, also known as core-binding factor alpha 1, is a master regulator of osteoblastic differentiation. Runx2 is an initial marker of osteoblastic differentiation of VSMCs [190]. RUNX2 expression was evidenced by its presence in CKD patients; Runx2 selectively exists in calcified arterial tissues [191,192]. In the development of calcification with atherogenic lesions, hydrogen peroxide triggers the phenotypic switch of VSMC due to increased *Runx2* expression and transactivation through Akt signaling. 

Inorganic phosphate, a major contributor to VC, induces *Runx 2* expression [193]. It acts as a transcription factor that increases expression of osteogenic genes (*OC, OPN, osterix, ALP, type-1 collagen*) [194]. In particular, *osterix* is a direct target of Runx2 and is absolutely required for the Runx2-mediated calcifying phenotype of VSMC.

##### Pyruvate Dehydrogenase Kinase (PDK)4

PDK, which consists of four isotypes, is a main regulator of the pyruvate dehydrogenase complex (PDC), a master regulator of cellular energy metabolism in mitochondria [195]. Among the four isotypes, the role of PDK4, in VC, by induction of osteogenic differentiation of VSMCs, was reported [196,197], while phosphorylation of PDHE1α was localized to calcified lesions in VC patients. In addition, high inorganic phosphate (Pi)-induces *PDK4* expression, and elicits phosphorylation of PDHE1α, in VSMCs, but not in PDKs 1, 2, or 3. PDK4^−/−^ mice showed decreased calcium deposition in ex vivo vessel ring models cultured with high Pi, with significant attenuation of calcium deposition challenged by vitamin D_3_ administration [197]. Moreover, pharmacological inhibition of PDK ameliorated calcification in VSMC and in animal models. Mechanistically, PDK4 interacted with Smads 1, 5, and 8, and induced osteogenic expression. Therefore, it was thought that inhibition of PDK4 may affect bone formation or bone mineral density, but surprisingly, there was no difference in bone mineral density in *PDK4*^−/−^ mice [197].

While PDK4 is an important protein present in mitochondria, and a well-known metabolism regulator, its role in VC still requires investigation [198]. To that end, the phenotype of *PDK4*^−/−^ mice, or pharmacological PDK inhibition, was explained by inhibition of osteoblastic transdifferentiation of VSMCs, and the prevention of mitochondrial dysfunction. However, phenomenologically significant calcification was reduced, indicating that other interdependent or independent mechanisms might be consequential [197,199]. In addition, recently, processes reported to associate with VC include mitochondrial dynamics, mitophagy, ROS, matrix vesicle release, apoptosis, and degradation of matrix [146,200,201,202]. Considering the results of animal experiments, further studies are needed, to determine whether these phenomena are related to PDK4.

#### 4.7.2. Inhibitor of VC

##### Osteoprotegerin (OPG)

Osteoprotegerin (OPG) binds to the RANKL, thereby interfering with RANK-RANKL interaction [203,204]. Inhibition of RANKL-RANK dimers inhibits the differentiation of osteoclast maturation, and thus, bone resorption [205,206,207]. Overexpression of OPG in mice shows reduced osteoclast differentiation from precursor cells, elevating bone mass. Therefore, OPG was initially considered as an efficient therapy to block RANKL, and retard osteoclast differentiation, in osteoporosis.

However, the role of OPG and RANKL in VC remains controversial [208,209]. In spite of the beneficial properties of OPG in experimental models, expression of OPG and RANKL are observed in atherosclerotic lesions of mice and humans. Additionally, circulating levels of OPG associate with increased carotid intima-media thickness, arterial stiffness, coronary artery disease, and cardiovascular risk factors (hyperlipidemia, diabetes, hypertension and metabolic syndrome) [210,211,212,213]. These seemingly contradictory observations suggest that, the increased number of OPG human clinical trials is more likely a result of compensatory response, for the prevention of further progression of vascular lesions [203,214]. While OPG neutralizes RANK/RANKL interactions, it is inefficient at high RANKL concentrations [215]. Indeed, at higher RANKL concentrations, monoclonal antibodies against RANKL (e.g., Denosumab) work more effectively than OPG, in terms of neutralizing RANKL. Taken together, despite many biological benefits related to OPG in VC prevention, its modulation and possible contribution are still uncertain. Nevertheless, its potential as a tool for predicting plaque vulnerability, as well as being a therapeutic target, warrants further investigation.

##### Osteopontin (OPN)

OPN is an extracellular phosphoprotein that has negatively charged phosphoserines. This feature provides strong affinity for hydroxyapatite [216]. OPN is present in mineralized tissues such as bones and teeth. OPN affects mineralization by preventing calcium crystal growth and accelerating osteoclast function. Increased levels of OPN are observed in calcified plaques; however, in healthy arteries, OPN is not present [217].

OPN is also found in inflammatory diseases such as high CV burden, left ventricle remodeling, and atherosclerotic lesions [218,219,220]. Its inhibitory effect on VC, through prevention of mineral crystal generation and crystal growth, has been reported [221,222,223]. In particular, phosphorylation of OPN is important for the inhibition of VC [224]. In support of this, dephosphorylated OPN is found in severe valvular calcification in patients with calcific aortic valve disease, suggesting that its phosphorylation is crucial as a protective role in ectopic calcification [225].

##### Matrix Gla Protein (MGP)

MGP is a vitamin K-dependent calcification inhibitor, highly expressed in calcium phosphate-deposited SMCs [226]. The role of MGP in calcification has been presented in several reports. For example, MGP-deficient mice are embryonic lethal, as a result of pathological arterial calcification [227]. On the other hand, MGP overexpression suppresses vascular BMP activity, atherosclerosis, intimal and medial calcification, and inflammation, in an atherosclerosis MGP transgenic mouse model [228]. MGP also binds to the crystal, thereby preventing its growth and inhibiting the osteoinductive effect of BMP2 [229,230]. It has been found that MGP has a different conformation by phosphorylation and carboxylation. Under pathological conditions, including inflammation, active MGP forms are necessary to counteract calcification [231,232].

Fully active MGP requires vitamin K-dependent carboxylation of glutamate, at positions 2, 37, 41, 48, and 52, and serine phosphorylation, at positions 3, 6, and 9, by a Golgi-casein kinase. Therefore, MGP has four conformations: dephospho-uncarboxylated MGP (dp-uc MGP), dephospho-carboxylated MGP (dp-uMGP), phosphorylated-uncarboxylated MGP (p-uc MGP), and phosphorylated-carboxylated MGP (p-cMGP). Released active MGP acts as a local inhibitor of calcification [233,234]. MGP is also ubiquitously expressed in multiple organs, and it is likely that this widespread expression of MGP has considerably more functions than are known for local inhibitor of calcification [235,236,237,238,239]. Recently, plasma dephospho-uncarboxylated MGP (dp-ucMGP) is used as a biomarker reflecting pool vitamin K status.

In vascular systems, stress increases MGP transcription in its uncarboxylated form. During calcifying conditions, insufficient vitamin K also increases production of total uncarboxylated MGP [233]. The p-uc MGP associates with the local VC, whereas dp-ucMGP has no or limited affinity to calcium. Thus, dp-ucMGP levels are higher than p-ucMGP in blood [240,241]. Circulating dp-ucMGP positively associates with the calcification score of the patient population [242], as supported by high dp-uc MGP levels in type 2 diabetes patients or arterial stiffness with high CVD risk, both of which represent diseases associated with calcification [243]. In addition, high dp-uc MGP levels reflect poor vitamin K status; therefore dp-uc MGP seems to be a promising tool in the assessment of CV risk, and prediction of VC [242].

##### Fetuin-A (Fet-A)

Fet-A has a strong inhibitory calcification property, and thus, a protective role against atherosclerosis [244]. It is well known that *Fet-A* knockout mice have prominent ectopic calcification [245]. Fet-A also has a capacity to bind early calcium phosphate crystals, and repress growth of crystal and mineral deposition [246]. Moreover, Fet-A containing calciprotein allows for the clearance of crystals, and reduction of the inflammatory response [247,248]. In this regard, circulating Fet-A is related to bone mass, and formation biomarkers, in contrary to bone resorption biomarkers.

Several studies report that VC, arterial stiffness, and increased risk of mortality and CV events, are related to low serum levels of Fet-A [249,250,251,252,253], which is also decreased in dialysis patients and negatively correlates with valvular calcification and abdominal aortic calcification [254,255]. Low protein diets, which are low in phosphate, promote VC by lowering Fet-A while increasingly precipitating calcium-phosphate complexes, in the serum of an experimental uremic rat model [256]. In VC, Fet-A is an inhibitor of calcification, which acts by protecting cells from crystal growth or deposition [257]. In this context, Fet-A is considered an important regulator of both bone resorption and VC.

##### Pyrophosphate (PPi)

Extracellular pyrophosphate (PPi) is synthesized from extracellular ATP, and prevents hydroxyapatite formation and growth [258]. It was first recognized as an endogenous inhibitor of biomineralization [259]. There are two major enzymes involved in pyrophosphate metabolism. The action of ENPP is to produce PPi, by hydrolysis of ATP, but NPP3 also promotes VC by hydrolyzing PPi to Pi [260,261,262]. Membrane protein ankyrin (progressive ankylosis or ANKH) regulates PPi levels, through release of intracellular PPi to the extracellular environment. [263]. Pyrophosphate is known as an inhibitor of nucleation of amphorous calcium and phosphate, and inhibits growth of hydroxyapatite and crystals through binding to the hydroxyapatite surface [259,264]. 5′-ectonucleotidase (5NT, also known as CD73) hydrolyzes ADP or AMP into adenosine [265]. Adenosine returns to the cell via equilibrative nucleotide transporter 1 (ENT1) [266,267] and is used for renewal of ATP [133].

As previously reported, diminished circulating PPi concentrations are common in VC [268]. TNAP, which hydrolyzes PPi to Pi, is the most studied enzyme involved in VC. Therefore, it makes sense that ectopic calcification occurs when TNAP is upregulated [269]. Moreover, overexpression of TNAP has also been observed in aortic calcification in CKD patients [270], a cause-and-effect event also supported by animal experiments [271]. For example, in one rat CKD model, calcium deposition leads to unexpected local increases in ANK expression, and late increases in ENPP1. These changes result in reduced plasma PPi concentrations as a later event [271]. In addition, *ENPP1* knockout mice have reduced levels of circulating PPi, accompanied by increased ectopic calcification [272]. As supporting data, intraperitoneal injection of PPi into the adenine-induced uremic calcification model showed a 70% decrease of calcium content [273]. In addition, orally administered PPi has also been shown to reduce calcification in other calcification models [274], thus reinforcing a protective role for PPi, in VC.

##### BMP7

In contrast to BMP2, BMP 7 has been proposed to play an inhibitory role in VC. For example, administration of BMP7 in animal models of CKD- mineral and bone disorder (CKD-MBD), with secondary hyperparathyroidism, provokes bone formation, and increases bone mass [176,275,276]. A protective role of BMP7, in VC development, is efficacious in *LDL* receptor-null mice with CKD [172,277], while *BMP7* reduced transformation to VSMC osteogenic phenotypes, and impaired VC [278].

BMP7 notably decreases expression of profibrotic and epithelial-to-mesenchymal transition (EMT)- related genes, and returns Pi levels to normal [279]. However, total calcium content did not change, in a severely calcified aorta, in response to BMP7 [280], suggesting that a VC-preventive role of BMP7 largely depends on the degree of calcification [277].

BMP7 also leads to lowering plasma phosphorus, in spite of persistent uremia and a high phosphate diet [275], while it also improves both high- and low-turnover bone diseases [276]. A previous report suggests that BMP7 begot no differences in trabecular number or spacing in uremic animals, but did affect osteoblast number, mineralizing surfaces, and bone formation rates [276].

## 5. Mitochondrial Dysfunction, and Defective Mitophagy, in Vascular Calcification

Mitochondria are organelles central to many cellular activities, including energy production, metabolism, and aging. In addition, they are implicated in response to apoptotic signals, in which mitochondria release cytochrome C, from the inner membrane to the cytosol, which in turn, induces a subsequent cascade of caspase activation and apoptosis [281]. This accompanies inhibition of ATP synthesis, mitochondrial depolarization, by mitochondrial permeability transition pore opening. Mitochondria in VSMCs are no exception, in which Pi-induced VSMC apoptosis is mediated by mitochondrial dysfunction [79].

In our own studies, we found that Pi decreases mitochondrial membrane potential, ATP synthesis, and increases caspase-3- and caspase-9-dependent apoptosis and calcification (Figure 3). In contrast, α-lipoic acid, one of the most potent antioxidants, prevented VSMC calcification by decreasing ROS and reversing mitochondrial dysfunction [79]. By contrast, apurinic/apyrimidinic endonuclease 1/redox factor-1 (APE1/Ref-1), which plays a pleiotropic role in the reduction of ROS, was impaired during Pi-induced VSMC calcification [282]. Similarly, hydrogen peroxide and resveratrol were also effective in preventing VC, by augmentation of nuclear factor erythroid 2-related factor 2 (Nrf2) signaling [283,284].

Recent biomedical literature highlights mitochondrial dynamics as a key feature linked to mitochondrial dysfunction or energy supply. For example, a nutrient-rich environment associates with a fragmented mitochondrial network, whereas starvation tends to elongate mitochondria [285]. This dynamic process, and especially, mitochondial fission, is increased in mitochondrial dysfunction [286]. This phenomenon is linked with a mitochondria quality-control program, in which the E3 ubiquitin ligase, parkin, is recruited to dysfunctional, fissed mitochondria, to ubiquitinate mitochondrial proteins for proteosomal degradation. This process promotes the engulfment of mitochondria by autophagosomes [287]. Mitochondrial fission is predominantly mediated by the dynamin-related GTPase dynamin related protein 1 (Drp1), whereas fusion is mediated by mitofusin (Mfn), and optic dominant atrophy 1 (Opa1). In the calcified human carotid artery, Drp1 expression is increased [288], while Drp1 knockdown attenuated human valve interstitial cell calcification, and retard VSMC migration. Pharmacological Drp1 inhibition ameliorated H_2_O_2_-induced mouse SMCs calcification [288]. This was further supported by a recent observation that another antioxidant quercetin, reverses mitochondrial dysfunction by inhibition of ROS formation, and mitochondrial fission, by inhibiting Drp1 [202] (Figure 3).

In clinical perspective, the mechanism by which high urea contributes to VC was recently revealed. High urea, as observed in CKD, causes a posttranslational modification known as carbamylation [289]. Intriguingly, mitochondrial proteins, including ATP synthase, are also carbamylated, deteriorating their own function, thereby inducing oxidative stress and mitochondrial dysfunction [289]. This, in turn, causes suppression of ENPP, the aforementioned inhibitor of ectopic calcification. As might be expected, calcification was prevented by a superoxide scavenger, underscoring the role of mitochondrial dysfunction and oxidative stress in VC in high urea environments, such as CKD [289]. More recent studies showed that MOTS-c, a mitochondrial-derived peptide, is effective in vitamin D_3_ and nicotine-induced VC [290].

Certain clinical situations, such as hypoxia, also promote calcification. For example, patients with bronchial asthma or chronic obstructive pulmonary disease are susceptible to arterial calcification [291,292]. Hypoxia-inducible factor 1α (HIF-1α), a transcription factor upregulated by hypoxia, can drive aerobic glycolysis. In addition to previous literature suggesting a critical role of HIF-1α in Pi-induced VC [293], another recent study demonstrated that hypoxia-dependent HIF-1α upregulates *Runx2*, thereby inducing calcification of mouse aorta [294]. These findings indirectly suggest a role of mitochondrial dysfunction in VC, given that HIF-1α is also upregulated in mitochondrial dysfunction.

A recent investigation supports this notion. In that work, exogenous lactate treatment accelerated VSMC calcification, along with impaired mitochondrial function, as evidenced by the opening rate of the mitochondrial permeability transition pore, depolarization of mitochondrial membrane potential, and downregulation of mitochondrial biogenesis markers. Intriguingly, lactate inhibited mitophagy, whereas BCL2-Interacting Protein 3 (BNIP3)-mediated mitophagy restored mitochondrial function, biogenesis, and reversed lactate-induced VSMC calcification [139] (Figure 3). The same authors identified a nuclear receptor, NR4A1, as an inhibitor of mitophagy, as well as a promoter of mitochondrial fission [295]. Given that lactate is a byproduct of aerobic glycolysis, these investigations collectively suggest an intertwined role of mitochondrial function-mitochondrial dynamics, and mitophagy, in the pathogenesis of VC.

## 6. Inflammation and Immune Dysregulation in the Pathogenesis of Vascular Calcification

CKD is a representative condition wherein inflammation and VC share pathogenesis. Indeed, a recent report claims that vascular media calcification and related inflammation exist even in the early stages of CKD [296]. It has also been reported that mRNA levels of components of the Nalp3 inflammasome complex, including *Nalp3, ASC*, and *caspase-1* were upregulated in calcifying VSMCs. Inflammatory cytokines, for example, interleukin-1 beta (IL-1β) stimulates VSMC calcification in vitro [78], whereas inhibition of inflammasome activation by Nalp3 KD reduced IL-1β secretion and inhibited VSMC calcification [78]. The following study also emphasizes the presence of IL-1β calcified coronary arteries in *ApoE*^−/−^ mice fed a HFD [297]. Here, the authors suggested Rac2 as a potential therapeutic target against VC. Therein, Rac2 prevented VC through its suppression of Rac1-dependent macrophage interleukin-1β (IL-1β) expression [297]. These findings highlight the importance of pro-inflammatory macrophages in VC progression. Furthermore, interleukin-6 (IL-6)/soluble interleukin-6 receptor (sIL-6R) complexes induced transformation of human VSMCs into an osteoblast phenotype, resulting in subsequent VC [298]. Tumor necrosis factor alpha (TNF-α), and IL-1β, induced the endothelial-to-mesenchymal transition in human primary aortic endothelial cells, thereby promoting them for BMP-9-mediated osteogenic differentiation. Here, downregulation of the BMP type II receptor, BMPR2, was required to enhance BMP-9-induced mineralization [299].

M0-like macrophages, upon exposure to calcium or phosphate nanocrystals, undergo M1-like polarization, which is evidenced by facilitated release of inducible nitric oxide synthase (iNOS) and TNF-α [300,301,302], suggesting recruitment of inflammatory immune cells to the site of calcification, occurs during calcifying events. Mechanistically, RANKL present in calcifying VSMC can promote migration and differentiation of macrophages into osteoclast-like cells in the atherosclerotic lesion, shaping crosstalk between VSMC and macrophage [303]. Of note, in the hepatocyte, treatment with serum IL-6, which is increased in CKD, resulted in an 8.9-fold downregulation of Fet-A gene expression [304]. As stated before, because Fet-A is a potent inhibitor of VC, it seems that besides its role in situ, the role of systemic macrophage-derived IL-6 is also indispensable for VC pathogenesis.

A recent study highlights the role of transient receptor potential canonical 3 (TRPC3) in macrophage in VC [305,306,307]. TRPC3 deficiency reduces ER stress-induced apoptosis exclusively in M1 macrophages [305]. Macrophage-specific TRPC3 deletion was sufficient to reduce atherosclerotic plaque in high fat diet-fed LDL receptor knockout VC rodent model. Furthermore, BMP2 as well as Runx2 expressing bone marrow derived macrophages was decreased in aortic root plaque [306]. This finding corroborates the importance of macrophage recruitment in VC. The more detailed and elegant two reviews are by Henaut et al. [301,302].

## 7. Conclusions

Despite a myriad of studies of how to overcome VC, the complexity and diversity of VC pathophysiology has impeded the discovery of the optimal drug targets, as well as drug development. In the past few decades, accumulating evidence has increased our knowledge of the pathogenesis of VC. That work has shown that beyond a high-phosphate environment, several key proteins, related to VC pathophysiology, have been identified (Figure 2). Furthermore, very recent evidence underscores the importance of dysfunctional organelles, from ER stress to autophagy impairment. A decisive role of perturbed mitochondrial dynamics, coupled to defective mitophagy, is being investigated. As a consequence, therapeutic strategies targeting inflammation or inflammatory immune cells are anticipated to reduce unmet clinical needs in VC.

## Figures and Tables

**Figure 1 ijms-21-02685-f001:**
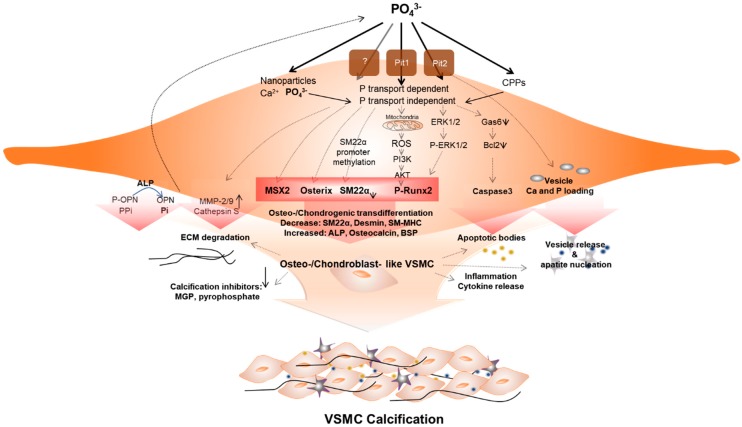
Vascular smooth muscle cells (VSMCs) under hyperphosphatemic conditions. Hyperphosphatemic milieu affects VSMC cellular fate by delivering phosphate (P) into VSMC via phosphate transport (Pit1 or Pit2) dependent, independent (as nanoparticles) or as a form of calciprotein particle (CPP). Followed by diverse signaling pathways that enhance sensitivity of VSMCs to calcification, phenotypic differentiation into osteogenic/chondroblast-like VSMCs occurs. This process includes signaling pathways that induce expression of the osteogenic transcription factors *Msh Homeobox 2* (MSX2), Osterix, runt-related transcription factor 2 (Runx2), and alkaline phosphatase (ALP). These changes acceleratedly reduce levels of calcification inhibitors. In addition, ROS generated by P-induced mitochondrial dysfunction activates Runx2 via phosphoinositide 3-kinase (PI3K)/protein kinase B (PKB or AKT) signaling and increased apoptosis promote apoptotic bodies or vesicle release. In addition, extracellular matrix (ECM) degradation and inflammatory cytokine releases are increased. These factors create pro-calcifying environment contributing to vascular calcification.

**Figure 2 ijms-21-02685-f002:**
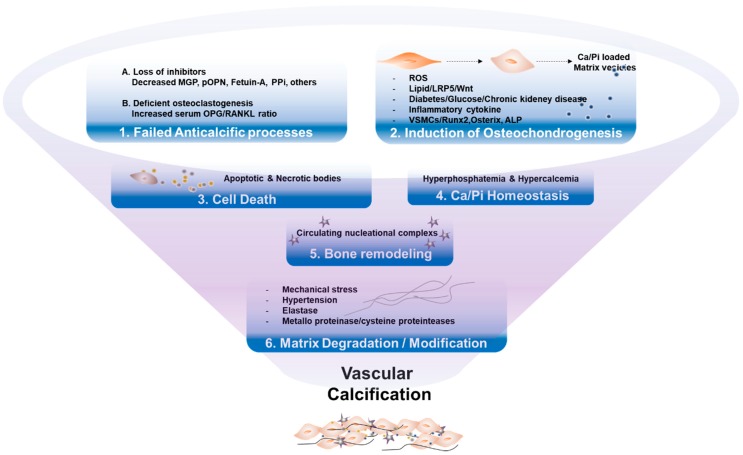
Key mechanisms of vascular calcification. (1) Failure of anti-calcification processes, due to loss of inhibitors and deficiency of constitutively expressed mineralization inhibitors, leads to vascular calcification. (2) Various stressors induce osteogenic transdifferentiation of VSMCs, products of matrix vesicles, which act as a nidus of calcium phosphate deposition. (3) Cell death by apoptosis or necrosis leads to release of apoptotic bodies, or necrotic debris, which may act as nucleation of apatite. (4) Abnormal mineral homeostasis causes deposits calcium phosphate hydroxyapatite. (5) Nucleational complexes formed during bone remodeling, promote ectopic mineralization. (6) Matrix degradation/modifications, caused by environmental stressors, are involved in vascular calcification.

**Figure 3 ijms-21-02685-f003:**
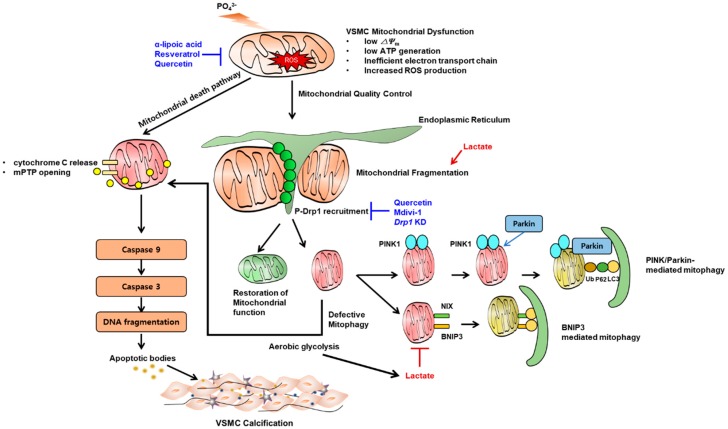
Key mechanism by which mitochondrial dysfunction contributes to vascular calcification. High phosphate induces vascular smooth muscle cell (VSMC) mitochondrial dysfunction represented by low mitochondrial membrane potential, low ATP synthesis and increased reactive oxygen species as a result of inefficient electron transport chain. This leads to release of mitochondrial permeability transition pore (mPTP) and release of cytochrome C which in turn, induces a subsequent cascade of caspase-9 and caspase-3 activation and apoptosis. Mitochondria require quality control in the presence of mitochondrial stress. The representative phenomenon is mitochondrial fission which is mainly mediated by phosphorylated Drp1 recruitment on the site of fission. Once fragmented, dysfunctional mitochondria undergo mitophagy (canonical PINK1-parkin mediated or BCL2-Interacting Protein 3 (BNIP3) mediated). When mitophagic clearance is defective, dysfunctional mitochondria undergo apoptosis. Dysfunctional mitochondria tend to rely on aereobic glycolysis rather than oxidative phosphorylation which might further produce lactate. Lactate promotes mitochondrial fission and blocks mitophagy, both of which promote apoptosis. Some chemicals are preventive in experimental models of vascular calcification (shown in blue text, also see main text).

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
