# Peer review of "Vascular Calcification—New Insights into Its Mechanism"

_ijms, 2020, doi:10.3390/ijms21082685_

Round 1
Reviewer 1 Report
The authors have deepened the single effect and role of molecules involved in the vascular calcification process. The paper is difficult to read and repetitive in some passages. I suggest to shorten the longer sentences, in order to make the paper more easily readable.
Furthermore, in my opinion, other issues should be BRIEFLY discussed:
- The authors reported the role of CKD and diabetes several times in the text. However, they did not stress the role of high blood pressure appropriately. Hypertension may cause arterial stiffening and vascular calcification through several mechanisms that the authors should take into account (PMID: 29981188; 15731494).
- The role of Lp(a) have recently emerged in cardiovascular disease and atherosclerotic process. The authors should take into account also this factor (PMID: 12196525; 31047003).
Minor issues:
- The figures should be cited in the text appropriately.
- The too many abbreviations make the paper difficult to read. Please, omit those superfluous or not immediately understandable (i.e. EV for Extracellular vesicles or MV for matrix vesicles…).
Author Response
Response Letter to Reviewer 1
First of all, we would like to express two reviewers and the editor for critical and informative comments. We faithfully addressed the issue in a point-by-point manner. We are willing to do another round of revision if our revised manuscript does not fulfill the reviewer’s requirement. Major changes that were made are in red text in the revised manuscript.
The authors have deepened the single effect and role of molecules involved in the vascular calcification process. The paper is difficult to read and repetitive in some passages. I suggest to shorten the longer sentences, in order to make the paper more easily readable.
Furthermore, in my opinion, other issues should be BRIEFLY discussed:
- The authors reported the role of CKD and diabetes several times in the text. However, they did not stress the role of high blood pressure appropriately. Hypertension may cause arterial stiffening and vascular calcification through several mechanisms that the authors should take into account (PMID: 29981188; 15731494).
Response) We deeply appreciate the reviewer’s considerate suggestions. According to the reviewer’s suggestion, we tried to re-write as concisely as possible to avoid repetitive & redundant sentences. We also briefly addressed the role of hypertension in arterial stiffness and vascular calcification [line 49~52, line 94-105].
- The role of Lp(a) have recently emerged in cardiovascular disease and atherosclerotic process. The authors should take into account also this factor (PMID: 12196525; 31047003).
Response) We appreciate reviewer’s insightful comment. The role of Lp(a) in the atherosclerosis was dealt in the revised paper section 4.6 [line 335-348 ].
Minor issues:
- The figures should be cited in the text appropriately.
Response) We appreciate reviewers for pointing out what we have critically missed.
Figure 1 is cited in line 145.
Figure 2 is cited in line 361 and 657
Lastly, Figure 3 is cited in line 591 and line 614.
- The too many abbreviations make the paper difficult to read. Please, omit those superfluous or not immediately understandable (i.e. EV for Extracellular vesicles or MV for matrix vesicles…).
Response) We agree with the reviewers that some words which do not require abbreviations were unnecessarily abbreviated, thus making the manuscript difficult to be understood. We removed dispensable abbreviations including EV, MV, ncRNA and etc.

Reviewer 2 Report
The review by Sun Joo Lee discusses the process of vascular calcification and its mechanism in detail. The review is well written and compiled by citing fifty percent of research articles published in past 5 years. They have successfully included types of vascular calcification (VC), its mechanism and the factors involved in the process of VC. However, the review could be improved by addressing the concerns below:
1) The authors should discuss briefly about valve calcification and calciphylaxis in their section 2 classification of vascular calcification, depending on site and etiology.
2) The authors should provide a table of all the abbreviations used in the review separately.
3) The authors should briefly compare and contrast micro and macrocalcification to give readers a better understanding in intimal calcification section.
4) The authors in line 76 have mentioned that osteoblast like cells differentiate into SMCs in medial calcification but it’s the other way around.
5) In the section 4.6 Osteogenic Markers in VC and Osteoporosis, it would be better if authors could separate the factors under subheadings of inducers and inhibitors of VC or they can provide a separate table in the same format.
6) Figure 2 is very similar to the figure 1 in the review by Cecilia M. Giachelli published in 2004 so please revise (https://jasn.asnjournals.org/content/15/12/2959)
7) In section 6 Inflammation and Immune dysregulation in pathogenesis of VC, the review would be benefitted by including findings from the 2 recent studies (link provided) which shows a direct role of macrophages in VC by their ability to produce osteogenic factors like BMP-2 and Runx2.
https://www.sciencedirect.com/science/article/abs/pii/S0006291X17314079
https://www.sciencedirect.com/science/article/abs/pii/S0021915017314442
8) There is some problem with citation 146, please check and correct.
9) Minor sentence formation, grammatical error and spell check required throughout.
Author Response
Response Letter to Reviewer 2
First of all, we would like to express two reviewers and the editor for critical and informative comments. We faithfully addressed the issue in a point-by-point manner. We are willing to do another round of revision if our revised manuscript does not fulfill the reviewer’s requirement. Major changes that were made are in red text in the revised manuscript.
Reviewer2
The review by Sun Joo Lee discusses the process of vascular calcification and its mechanism in detail. The review is well written and compiled by citing fifty percent of research articles published in past 5 years. They have successfully included types of vascular calcification (VC), its mechanism and the factors involved in the process of VC. However, the review could be improved by addressing the concerns below:
1) The authors should discuss briefly about valve calcification and calciphylaxis in their section 2 classification of vascular calcification, depending on site and etiology.
Response) We appreciate the reviewer for pointing this out. We added this issue in section 2 according to the reviewer’s suggestion in the revised manuscript [line 107 -129 ]
2) The authors should provide a table of all the abbreviations used in the review separately.
Response) According to the reviewer’s comment and IJMS journal format, we added a list of abbreviation in alphabetical order in the last part of the revised manuscript [line 710-759 ]
3) The authors should briefly compare and contrast micro and macrocalcification to give readers a better understanding in intimal calcification section.
Response) This issue is briefly discussed in intimal calcification section as the reviewer has suggested [line 69-72].
4) The authors in line 76 have mentioned that osteoblast like cells differentiate into SMCs in medial calcification but it’s the other way around.
Response) We apologize for this terrible mistake. It was changed to “smooth muscle cells (SMCs) into osteoblast-like cells” in the revised manuscript [line 84].
5) In the section 4.6 Osteogenic Markers in VC and Osteoporosis, it would be better if authors could separate the factors under subheadings of inducers and inhibitors of VC or they can provide a separate table in the same format.
Response) We greatly appreciate the reviewer’s insightful comment. Among twelve markers, six of them were placed inducers of VC and the others were categorized as inhibitors of VC [line 363 -560].
6) Figure 2 is very similar to the figure 1 in the review by Cecilia M. Giachelli published in 2004 so please revise (https://jasn.asnjournals.org/content/15/12/2959)
Response) We made substantial changes in Figure 2. We appreciate the reviewer for pointing out this critical issue. We hope revised figure raises no concerns of plagarism.
7) In section 6 Inflammation and Immune dysregulation in pathogenesis of VC, the review would be benefitted by including findings from the 2 recent studies (link provided) which shows a direct role of macrophages in VC by their ability to produce osteogenic factors like BMP-2 and Runx2.
https://www.sciencedirect.com/science/article/abs/pii/S0006291X17314079
https://www.sciencedirect.com/science/article/abs/pii/S0021915017314442
Response) We were indeed not aware of these recent studies. We appreciate the reviewer for this valuable suggestion, and included in the revised manuscript [line 645-650].
8) There is some problem with citation 146, please check and correct.
Response) There was unexpected technical error which we have not recognized. We amended the reference.
9) Minor sentence formation, grammatical error and spell check required throughout.
Response) We did another round of English editing after revising the manuscript. We hope the revised manuscript is more easily readable.
